# Probabilistic Low-Rank Matrix Completion with Adaptive Spectral Regularization Algorithms

**Adrien Todeschini**
INRIA - IMB - Univ. Bordeaux
33405 Talence, France
Adrien.Todeschini@inria.fr

**François Caron**
Univ. Oxford, Dept. of Statistics
Oxford, OX1 3TG, UK
Caron@stats.ox.ac.uk

**Marie Chavent**
Univ. Bordeaux - IMB - INRIA
33000 Bordeaux, France
Marie.Chavent@u-bordeaux2.fr

## Abstract

We propose a novel class of algorithms for low rank matrix completion. Our approach builds on novel penalty functions on the singular values of the low rank matrix. By exploiting a mixture model representation of this penalty, we show that a suitably chosen set of latent variables enables to derive an Expectation-Maximization algorithm to obtain a Maximum A Posteriori estimate of the completed low rank matrix. The resulting algorithm is an iterative soft-thresholded algorithm which iteratively adapts the shrinkage coefficients associated to the singular values. The algorithm is simple to implement and can scale to large matrices. We provide numerical comparisons between our approach and recent alternatives showing the interest of the proposed approach for low rank matrix completion.

## 1 Introduction

Matrix completion has attracted a lot of attention over the past few years. The objective is to "complete" a matrix of potentially large dimension based on a small (and potentially noisy) subset of its entries [1, 2, 3]. One popular application is to build automatic recommender systems, where the rows correspond to users, the columns to items and entries may be ratings or binary (like/dislike). The objective is then to predict user preferences from a subset of the entries.

In many cases, it is reasonable to assume that the unknown $m \times n$ matrix $Z$ can be approximated by a matrix of low rank $Z \simeq AB^T$ where $A$ and $B$ are respectively of size $m \times k$ and $n \times k$, with $k \ll \min(m, n)$. In the recommender system application, the low rank assumption is sensible as it is commonly believed that only a few factors contribute to user's preferences. The low rank structure thus implies some sort of collaboration between the different users/items [4].

We typically observe a noisy version $X_{ij}$ of some entries $(i, j) \in \Omega$ where $\Omega \subset \{1, \ldots, m\} \times \{1, \ldots, n\}$. For $(i, j) \in \Omega$

$$X_{ij} = Z_{ij} + \varepsilon_{ij}, \varepsilon_{ij} \stackrel{\text{iid}}{\sim} \mathcal{N}(0, \sigma^2) \tag{1}$$

where $\sigma^2 > 0$ and $\mathcal{N}(\mu, \sigma^2)$ is the normal distribution of mean $\mu$ and variance $\sigma^2$. Low rank matrix completion can be adressed by solving the following optimization problem

$$\underset{Z}{\text{minimize}} \quad \frac{1}{2\sigma^2} \sum_{(i,j) \in \Omega} (X_{ij} - Z_{ij})^2 + \lambda \operatorname{rank}(Z) \tag{2}$$

where $\lambda > 0$ is some regularization parameter. For general subsets $\Omega$, the optimization problem (2) is computationally hard and many authors have advocated the use of a convex relaxation of (2) [5, 6, 4], yielding the following convex optimization problem

$$\underset{Z}{\text{minimize}} \quad \frac{1}{2\sigma^2} \sum_{(i,j)\in\Omega} (X_{ij} - Z_{ij})^2 + \lambda \|Z\|_* \tag{3}$$

where $\|Z\|_*$ is the nuclear norm of $Z$, or the sum of the singular values of $Z$. [4] proposed an iterative algorithm, called Soft-Impute, for solving the nuclear norm regularized minimization (3).

In this paper, we show that the solution to the objective function (3) can be interpreted as a Maximum A Posteriori (MAP) estimate when assuming that the singular values of $Z$ are independently and identically drawn (iid) from an exponential distribution with rate $\lambda$. Using this Bayesian interpretation, we propose alternative concave penalties to the nuclear norm, obtained by considering that the singular values are iid from a mixture of exponential distributions. We show that this class of penalties bridges the gap between the nuclear norm and the rank penalty, and that a simple Expectation-Maximization (EM) algorithm can be derived to obtain MAP estimates. The resulting algorithm iteratively adapts the shrinkage coefficients associated to the singular values. It can be seen as the equivalent for matrices of reweighted $\ell_1$ algorithms [6] for multivariate linear regression. Interestingly, we show that the Soft-Impute algorithm of [4] is obtained as a particular case. We also discuss the extension of our algorithms to binary matrices, building on the same seed of ideas, in the supplementary material. Finally, we provide some empirical evidence of the interest of the proposed approach on simulated and real data.

## 2  Complete matrix $X$

Consider first that we observe the complete matrix $X$ of size $m \times n$. Let $r = \min(m, n)$. We consider the following convex optimization problem

$$\underset{Z}{\text{minimize}} \quad \frac{1}{2\sigma^2} \|X - Z\|_F^2 + \lambda \|Z\|_* \tag{4}$$

where $\|\cdot\|_F$ is the Frobenius norm. The solution to Eq. (4) in the complete case is a soft-thresholded singular value decomposition (SVD) of $X$ [7, 4], i.e.

$$\widehat{Z} = \mathbf{S}_{\lambda\sigma^2}(X)$$

where $\mathbf{S}_\lambda(X) = \widetilde{U}\widetilde{D}_\lambda\widetilde{V}^T$ with $\widetilde{D}_\lambda = \text{diag}((\widetilde{d}_1 - \lambda)_+, \ldots, (\widetilde{d}_r - \lambda)_+)$ and $t_+ = \max(t, 0)$. $X = \widetilde{U}\widetilde{D}\widetilde{V}^T$ is the singular value decomposition of $X$ with $\widetilde{D} = \text{diag}(\widetilde{d}_1, \ldots, \widetilde{d}_r)$.

The solution $\widehat{Z}$ to the optimization problem (4) can be interpreted as the Maximum A Posteriori estimate under the likelihood (1) and prior

$$p(Z) \propto \exp\left(-\lambda \|Z\|_*\right)$$

Assuming $Z = UDV^T$, with $D = \text{diag}(d_1, d_2, \ldots, d_r)$ this can be further decomposed as

$$p(Z) = p(U)p(V)p(D)$$

where we assume a uniform Haar prior distribution on the unitary matrices $U$ and $V$, and exponential priors on the singular values $d_i$, hence

$$p(d_1, \ldots, d_r) = \prod_{i=1}^{r} \text{Exp}\left(d_i; \lambda\right) \tag{5}$$

where $\text{Exp}(x; \lambda) = \lambda \exp(-\lambda x)$ is the probability density function (pdf) of the exponential distribution of parameter $\lambda$ evaluated at $x$. The exponential distribution has a mode at 0, hence favoring sparse solutions.

We propose here alternative penalty/prior distributions, that bridge the gap between the rank and the nuclear norm penalties. Our penalties are based on hierarchical Bayes constructions and the related optimization problems to obtain MAP estimates can be solved by using an EM algorithm.

### 2.1  Hierarchical adaptive spectral penalty

We consider the following hierarchical prior for the low rank matrix $Z$. We still assume that $Z = UDV^T$, where the unitary matrices $U$ and $V$ are assigned uniform priors and $D = \text{diag}(d_1, \ldots, d_r)$. We now assume that each singular value $d_i$ has its own regularization parameter $\gamma_i$.

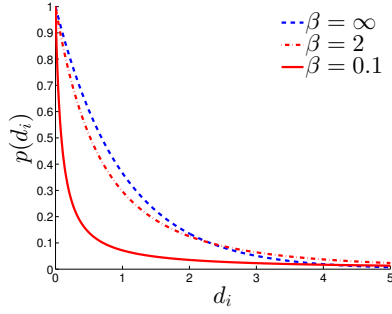

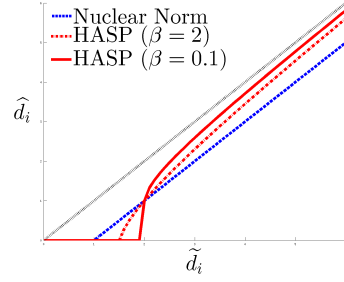

Figure 1: Marginal distribution $p(d_i)$ with $a = b = \beta$ for different values of the parameter $\beta$. The distribution becomes more concentrated around zero with heavier tails as $\beta$ decreases. The case $\beta \to \infty$ corresponds to an exponential distribution with unit rate.

Figure 2: Thresholding rules on the singular values $\widetilde{d}_i$ of $X$ for the soft thresholding rule ($\lambda = 1$), and hierarchical adaptive soft thresholding algorithm with $a = b = \beta$.

$$p(d_1, \ldots, d_r | \gamma_1, \ldots \gamma_r) = \prod_{i=1}^{r} p(d_i | \gamma_i) = \prod_{i=1}^{r} \text{Exp}(d_i; \gamma_i)$$

We further assume that the regularization parameters are themselves iid from a gamma distribution

$$p(\gamma_1, \ldots, \gamma_r) = \prod_{i=1}^{r} p(\gamma_i) = \prod_{i=1}^{r} \text{Gamma}(\gamma_i; a, b)$$

where $\text{Gamma}(\gamma_i; a, b)$ is the pdf of the gamma distribution of parameters $a > 0$ and $b > 0$ evaluated at $\gamma_i$. The marginal distribution over $d_i$ is thus a continuous mixture of exponential distributions

$$p(d_i) = \int_0^{\infty} \text{Exp}(d_i; \gamma_i) \, \text{Gamma}(\gamma_i; a, b) d\gamma_i = \frac{ab^a}{(d_i + b)^{a+1}} \tag{6}$$

It is a Pareto distribution which has heavier tails than the exponential distribution. Figure 1 shows the marginal distribution $p(d_i)$ for $a = b = \beta$. The lower $\beta$, the heavier the tails of the distribution. When $\beta \to \infty$, one recovers the exponential distribution of unit rate parameter. Let

$$pen(Z) = -\log p(Z) = -\sum_{i=1}^{r} \log(p(d_i)) = C_1 + \sum_{i=1}^{r} (a+1) \log(b + d_i) \tag{7}$$

be the penalty induced by the prior $p(Z)$. We call the penalty (7) the Hierarchical Adaptive Spectral Penalty (HASP). On Figure 3 (top) are represented the balls of constant penalties for a symmetric $2 \times 2$ matrix, for the HASP, nuclear norm and rank penalties. When the matrix is assumed to be diagonal, one recovers respectively the lasso, hierarchical adaptive lasso (HAL) [6, 8] and $\ell_0$ penalties, as shown on Figure 3 (bottom).

The penalty (7) admits as special cases the nuclear norm penalty $\lambda ||Z||_*$ when $a = \lambda b$ and $b \to \infty$. Another closely related penalty is the log-det heuristic [5, 9] penalty, defined for a square matrix $Z$ by $\log \det(Z + \delta I)$ where $\delta$ is some small regularization constant. Both penalties agree on square matrices when $a = b = 0$ and $\delta = 0$.

## 2.2  EM algorithm for MAP estimation

Using the exponential mixture representation (6), we now show how to derive an EM algorithm [10] to obtain a MAP estimate

$$\widehat{Z} = \arg \max_{Z} \left[ \log p(X|Z) + \log p(Z) \right]$$

i.e. to minimize

$$L(Z) = \frac{1}{2\sigma^2} \|X - Z\|_F^2 + \sum_{i=1}^{r} (a+1) \log(b + d_i) \tag{8}$$

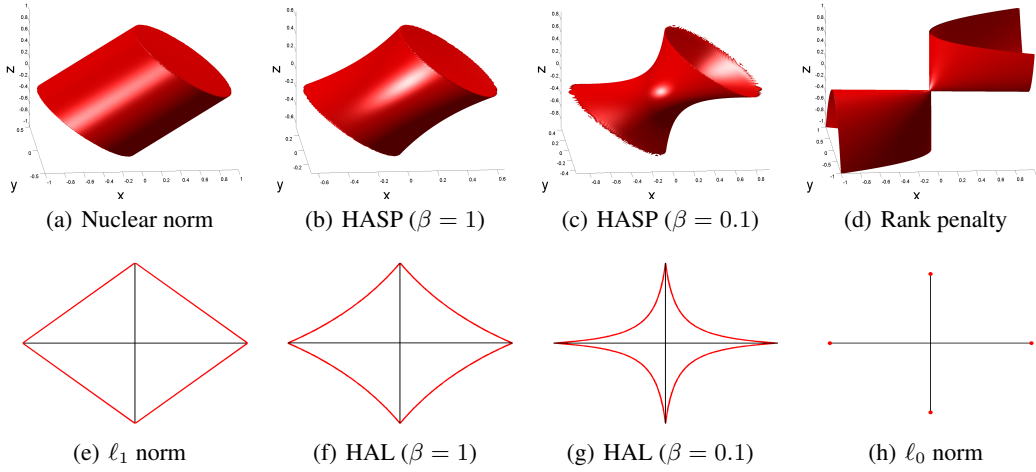

(a) Nuclear norm  (b) HASP ($\beta = 1$)  (c) HASP ($\beta = 0.1$)  (d) Rank penalty

(e) $\ell_1$ norm  (f) HAL ($\beta = 1$)  (g) HAL ($\beta = 0.1$)  (h) $\ell_0$ norm

Figure 3: Top: Manifold of constant penalty, for a symmetric $2 \times 2$ matrix $Z = [x, y; y, z]$ for (a) the nuclear norm, (b-c) hierarchical adaptive spectral penalty with $a = b = \beta$ (b) $\beta = 1$ and (c) $\beta = 0.1$, and (d) the rank penalty. Bottom: contour of constant penalty for a diagonal matrix $[x, 0; 0, z]$, where one recovers the classical (e) lasso, (f-g) hierarchical lasso and (h) $\ell_0$ penalties.

We use the parameters $\gamma = (\gamma_1, \ldots, \gamma_r)$ as latent variables in the EM algorithm. The E step is obtained by

$$Q(Z, Z^*) = \mathbb{E}\left[\log(p(X, Z, \gamma))|Z^*, X\right] = C_2 - \frac{1}{2\sigma^2}\|X - Z\|_F^2 - \sum_{i=1}^{r} \mathbb{E}[\gamma_i | d_i^*] d_i$$

Hence at each iteration of the EM algorithm, the M step consists in solving the optimization problem

$$\underset{Z}{\text{minimize}} \quad \frac{1}{2\sigma^2}\|X - Z\|_F^2 + \sum_{i=1}^{r} \omega_i d_i \tag{9}$$

where $\omega_i = \mathbb{E}[\gamma_i | d_i^*] = \frac{\partial}{\partial d_i^*}\left[-\log p(d_i^*)\right] = \frac{a+1}{b+d_i^*}$.

(9) is an adaptive nuclear norm regularized optimization problem, with weights $\omega_i$. Without loss of generality, assume that $d_1^* \geq d_2^* \geq \ldots \geq d_r^*$. It implies that

$$0 \leq \omega_1 \leq \omega_2 \leq \ldots \leq \omega_r \tag{10}$$

The above weights will therefore penalize less heavily higher singular values, hence reducing bias. As shown by [11, 12], a global optimal solution to Eq. (9) under the order constraint (10) is given by a weighted soft-thresholded SVD

$$\widehat{Z} = \mathbf{S}_{\sigma^2 \omega}(X) \tag{11}$$

where $\mathbf{S}_\omega(X) = \widetilde{U}\widetilde{D}_\omega\widetilde{V}^T$ with $\widetilde{D}_\omega = \text{diag}((\widetilde{d}_1 - \omega_1)_+, \ldots, (\widetilde{d}_r - \omega_r)_+)$. $X = \widetilde{U}\widetilde{D}\widetilde{V}^T$ is the SVD of $X$ with $\widetilde{D} = \text{diag}(\widetilde{d}_1, \ldots, \widetilde{d}_r)$ and $\widetilde{d}_1 \geq \widetilde{d}_2 \ldots \geq \widetilde{d}_r$.

Algorithm 1 summarizes the Hierarchical Adaptive Soft Thresholded (HAST) procedure to converge to a local minimum of the objective (8). This algorithm admits the soft-thresholded SVD operator as a special case when $a = b\lambda$ and $b = \beta \to \infty$. Figure 2 shows the thresholding rule applied to the singular values of $X$ for the HAST algorithm ($a = b = \beta$, with $\beta = 2$ and $\beta = 0.1$) and the soft-thresholded SVD for $\lambda = 1$. The bias term, which is equal to $\lambda$ for the nuclear norm, goes to zero as $\widetilde{d}_i$ goes to infinity.

**Setting of the hyperparameters and initialization of the EM algorithm**  In the experiments, we have set $b = \beta$ and $a = \lambda\beta$ where $\lambda$ and $\beta$ are tuning parameters that can be chosen by cross-validation. As $\lambda$ is the mean value of the regularization parameter $\gamma_i$, we initialize the algorithm with the soft thresholded SVD with parameter $\sigma^2\lambda$. It is possible to estimate the hyperparameter $\sigma$ within the EM algorithm as described in the supplementary material. In our experiments, we have found the results not very sensitive to the setting of $\sigma$, and set it to 1.

**Algorithm 1** Hierarchical Adaptive Soft Thresholded (HAST) algorithm for low rank estimation of complete matrices

Initialize $Z^{(0)}$. At iteration $t \geq 1$

- For $i = 1, \ldots, r$, compute the weights $\omega_i^{(t)} = \frac{a+1}{b+d_i^{(t-1)}}$
- Set $Z^{(t)} = \mathbf{S}_{\sigma^2 \omega^{(t)}}(X)$
- If $\frac{L(Z^{(t-1)}) - L(Z^{(t)})}{L(Z^{(t-1)})} < \varepsilon$ then return $\widehat{Z} = Z^{(t)}$

## 3 Matrix completion

We now show how the EM algorithm derived in the previous section can be adapted to the case where only a subset of the entries is observed. It relies on imputing missing values, similarly to the EM algorithm for SVD with missing data, see e.g. [10, 13].

Consider that only a subset $\Omega \subset \{1, \ldots, m\} \times \{1, \ldots, n\}$ of the entries of the matrix $X$ is observed. Similarly to [7], we introduce the operator $P_\Omega(X)$ and its complementary $P_\Omega^\perp(X)$

$$
P_\Omega(X)(i,j) = \begin{cases} X_{ij} & \text{if } (i,j) \in \Omega \\ 0 & \text{otherwise} \end{cases} \quad \text{and} \quad P_\Omega^\perp(X)(i,j) = \begin{cases} 0 & \text{if } (i,j) \in \Omega \\ X_{ij} & \text{otherwise} \end{cases}
$$

Assuming the same prior (6), the MAP estimate is obtained by minimizing

$$
L(Z) = \frac{1}{2\sigma^2} \|P_\Omega(X) - P_\Omega(Z)\|_F^2 + (a+1) \sum_{i=1}^r \log(b + d_i) \tag{12}
$$

We will now derive the EM algorithm, by using latent variables $\gamma$ and $P_\Omega^\perp(X)$. The E step is given by (details in supplementary material)

$$
\begin{aligned}
Q(Z, Z^*) &= \mathbb{E}\left[\log(p(P_\Omega(X), P_\Omega^\perp(X), Z, \gamma)) | Z^*, P_\Omega(X)\right] \\
&= C_4 - \frac{1}{2\sigma^2}\left\{\left\|P_\Omega(X) + P_\Omega^\perp(Z^*) - Z\right\|_F^2\right\} - \sum_{i=1}^r \mathbb{E}[\gamma_i | d_i^*] d_i
\end{aligned}
$$

Hence at each iteration of the algorithm, one needs to minimize

$$
\frac{1}{2\sigma^2} \|X^* - Z\|_F^2 + \sum_{i=1}^r \omega_i d_i \tag{13}
$$

where $\omega_i = \mathbb{E}[\gamma_i | d_i^*]$ and $X^* = P_\Omega(X) + P_\Omega^\perp(Z^*)$ is the observed matrix, completed with entries in $Z^*$. We now have a complete matrix problem. As mentioned in the previous section, the minimum of (13) is obtained with a weighted soft-thresholded SVD. Algorithm 2 provides the resulting iterative procedure for matrix completion with the hierarchical adaptive spectral penalty.

**Algorithm 2** Hierarchical Adaptive Soft Impute (HASI) algorithm for matrix completion

Initialize $Z^{(0)}$. At iteration $t \geq 1$

- For $i = 1, \ldots, r$, compute the weights $\omega_i^{(t)} = \frac{a+1}{b+d_i^{(t-1)}}$
- Set $Z^{(t)} = \mathbf{S}_{\sigma^2 \omega^{(t)}}\left(P_\Omega(X) + P_\Omega^\perp(Z^{(t-1)})\right)$
- If $\frac{L(Z^{(t-1)}) - L(Z^{(t)})}{L(Z^{(t-1)})} < \varepsilon$ then return $\widehat{Z} = Z^{(t)}$

**Related algorithms** Algorithm 2 admits the Soft-Impute algorithm of [4] as a special case when $a = \lambda b$ and $b = \beta \to \infty$. In this case, one obtains at each iteration $\omega_i^{(t)} = \lambda$ for all $i$. On the contrary, when $\beta < \infty$, our algorithm adaptively updates the weights so that to penalize less heavily higher singular values. Some authors have proposed related one-step adaptive spectral penalty algorithms [14, 11, 12]. However, in these procedures, the weights have to be chosen by some procedure whereas in our case they are iteratively adapted.

**Initialization** The objective function (12) is in general not convex and different initializations may lead to different modes. As in the complete case, we suggest to set $a = \lambda b$ and $b = \beta$ and to initialize the algorithm with the Soft-Impute algorithm with regularization parameter $\sigma^2 \lambda$.

**Scaling**   Similarly to the Soft-Impute algorithm, the computationally demanding part of Algorithm 2 is $\mathbf{S}_{\sigma^2 \omega^{(t)}}\left(P_\Omega(X) + P_\Omega^\perp(Z^{(t-1)})\right)$ which requires calculating a low rank truncated SVD. For large matrices, one can resort to the PROPACK algorithm [15, 16] as described in [4]. This sophisticated linear algebra algorithm can efficiently compute the truncated SVD of the "sparse + low rank" matrix

$$P_\Omega(X) + P_\Omega^\perp(Z^{(t-1)}) = \underbrace{P_\Omega(X) - P_\Omega(Z^{(t-1)})}_{\text{sparse}} + \underbrace{Z^{(t-1)}}_{\text{low rank}}$$

and can thus handle large matrices, as shown in [4].

# 4   Experiments

## 4.1   Simulated data

We first evaluate the performance of the proposed approach on simulated data. Our simulation setting is similar to that of [4]. We generate Gaussian matrices $A$ and $B$ respectively of size $m \times q$ and $n \times q$, $q \leq r$ so that the matrix $Z = AB^T$ is of low rank $q$. A Gaussian noise of variance $\sigma^2$ is then added to the entries of $Z$ to obtain the matrix $X$. The signal to noise ratio is defined as $\text{SNR} = \sqrt{\frac{\text{var}(Z)}{\sigma^2}}$. We set $m = n = 100$ and $\sigma = 1$. We run all the algorithms with a precision $\epsilon = 10^{-9}$ and a maximum number of $t_{max} = 200$ iterations (initialization included for HASI). We compute $err$, the relative error between the estimated matrix $\widehat{Z}$ and the true matrix $Z$ in the complete case, and $err_{\Omega^\perp}$ in the incomplete case, where

$$err = \frac{||\widehat{Z} - Z||_F^2}{||Z||_F^2} \quad \text{and} \quad err_{\Omega^\perp} = \frac{||\widehat{P}_\Omega^\perp(\widehat{Z}) - P_\Omega^\perp(Z)||_F^2}{||P_\Omega^\perp(Z)||_F^2}$$

For the HASP penalty, we set $a = \lambda\beta$ and $b = \beta$. We compute the solutions over a grid of 50 values of the regularization parameter $\lambda$ linearly spaced from $\lambda_0$ to 0, where $\lambda_0 = ||P_\Omega(X)||_2$ is the largest singular value of the input matrix $X$, padded with zeros. This is done for three different values $\beta = 1, 10, 100$. We use the same grid to obtain the regularization path for the other algorithms.

**Complete case**   We first consider that the observed matrix is complete, with $\text{SNR} = 1$ and $q = 10$. The HAST algorithm 1 is compared to the soft thresholded (ST) and hard thresholded (HT) SVD. Results are reported in Figure 4(a). The HASP penalty provides a bridge/tradeoff between the nuclear norm and the rank penalty. For example, value of $\beta = 10$ show a minimum at the true rank $q = 10$ as HT, but with a lower error when the rank is overestimated.

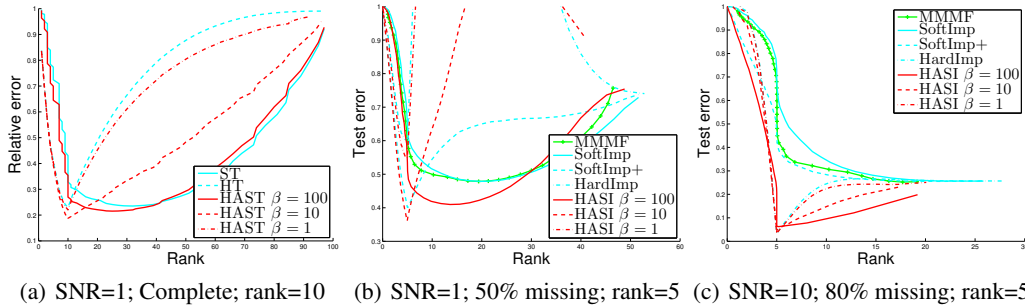

(a) SNR=1; Complete; rank=10   (b) SNR=1; 50% missing; rank=5   (c) SNR=10; 80% missing; rank=5

Figure 4: Test error w.r.t. the rank obtained by varying the value of the regularization parameter $\lambda$. Results on simulated data are given for (a) complete matrix with SNR=1 (b) 50% missing and SNR=1 and (c) 80% missing and SNR=10.

**Incomplete case**   Then we consider the matrix completion problem, and remove uniformly a given percentage of the entries in $X$. We compare the HASI algorithm to the Soft-Impute, Soft-Impute+ and Hard-Impute algorithms of [4] and to the MMMF algorithm of [17]. Results, averaged over 50 replications, are reported in Figures 4(b-c) for a true rank $q = 5$, (b) 50% of missing data and

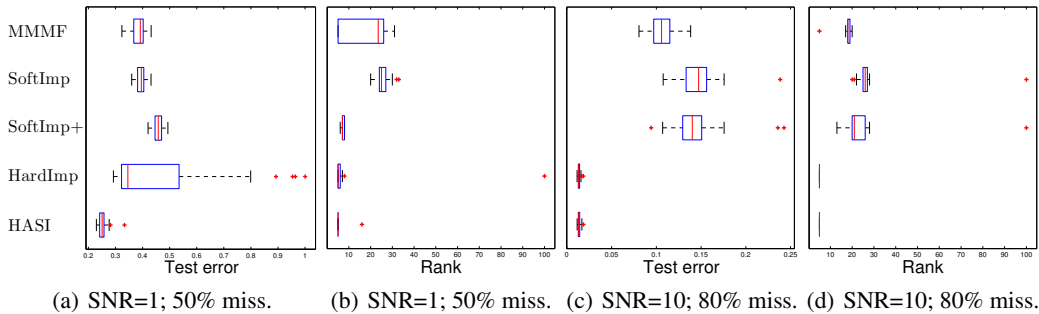

Figure 5: Boxplots of the test error and ranks obtained over 50 replications on simulated data.

Table 1: Results on the Jester and MovieLens datasets

| Method | Jester 1 $24983 \times 100$ $27.5\%$ miss. | | Jester 2 $23500 \times 100$ $27.3\%$ miss. | | Jester 3 $24938 \times 100$ $75.3\%$ miss. | | MovieLens 100k $943 \times 1682$ $93.7\%$ miss. | | MovieLens 1M $6040 \times 3952$ $95.8\%$ miss. | |
|---|---|---|---|---|---|---|---|---|---|---|
| | NMAE | Rank | NMAE | Rank | NMAE | Rank | NMAE | Rank | NMAE | Rank |
| MMMF | 0.161 | 95 | 0.162 | 96 | 0.183 | 58 | 0.195 | 50 | **0.169** | 30 |
| Soft Imp | 0.161 | 100 | 0.162 | 100 | 0.184 | 78 | 0.197 | 156 | 0.176 | 30 |
| Soft Imp+ | 0.169 | 14 | 0.171 | 11 | 0.184 | 33 | 0.197 | 108 | 0.189 | 30 |
| Hard Imp | 0.158 | 7 | 0.159 | 6 | 0.181 | 4 | 0.190 | 7 | 0.175 | 8 |
| HASI | **0.153** | 100 | **0.153** | 100 | **0.174** | 30 | **0.187** | 35 | 0.172 | 27 |

SNR $= 1$ and (c) $80\%$ of missing data and SNR $= 10$. Similar behavior is observed, with the HASI algorithm attaining a minimum at the true rank $q = 5$. We then conduct the same experiments, but remove $20\%$ of the observed entries as a validation set to estimate the regularization parameters $(\lambda, \beta)$ for HASI, and $\lambda$ for the other methods. We estimate $Z$ on the whole observed matrix, and use the unobserved entries as a test set. Results on the test error and estimated ranks over 50 replications are reported in Figure 5. For $50\%$ missing data, HASI is shown to outperform the other methods. For $80\%$ missing data, HASI and Hard Impute provide the best performances. In both cases, it is able to recover very accurately the true rank of the matrix.

## 4.2 Collaborative filtering examples

We now compare the different methods on several benchmark datasets. We first consider the Jester datasets [18]. The three datasets[1] contain one hundred jokes, with user ratings between -10 and +10. We randomly select two ratings per user as a test set, and two other ratings per user as a validation set to select the parameters $\lambda$ and $\beta$. The results are computed over four values $\beta = 1000, 100, 10, 1$. We compare the results of the different methods with the Normalized Mean Absolute Error (NMAE)

$$\text{NMAE} = \frac{\frac{1}{card(\Omega_{test})} \sum_{(i,j) \in \Omega_{test}} |X_{ij} - \widehat{Z}_{ij}|}{\max(X) - \min(X)}$$

where $\Omega_{test}$ is the test set. The mean number of iterations for Soft-Impute, Hard-Impute and HASI (initialization included) algorithms are respectively 9, 76 and 76. Computations for the HASI algorithm take approximately 5 hours on a standard computer. The results, averaged over 10 replications (with almost no variability observed), are presented in Table 1. The HASI algorithm provides very good performance on the different Jester datasets, with lower NMAE than the other methods.

Figure 6 shows the NMAE in function of the rank. Low values of $\beta$ exhibit a bimodal behavior with two modes at low rank and full rank. High value $\beta = 1000$ is unimodal and outperforms Soft-Impute at any particular rank.

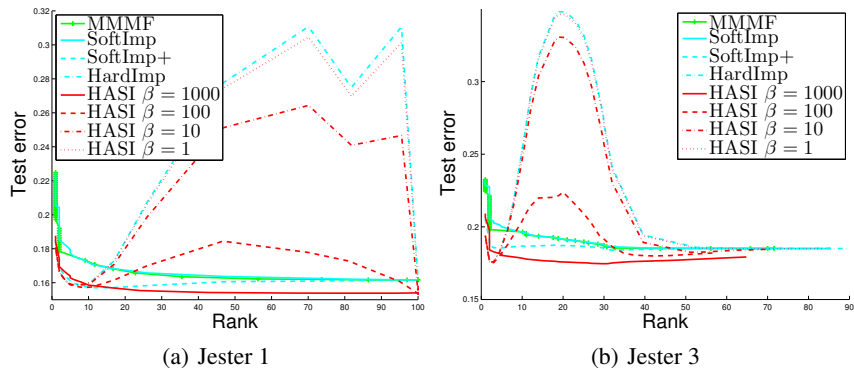

|           |           |
|:---------:|:---------:|
| (a) Jester 1 | (b) Jester 3 |

Figure 6: NMAE on the test set of the (a) Jester 1 and (b) Jester 3 datasets w.r.t. the rank obtained by varying the value of the regularization parameter $\lambda$. The curves obtained on the Jester 2 dataset are hardly distinguishable from (a) and hence are not displayed due to space limitation.

Second, we conducted the same comparison on two MovieLens datasets[2], which contain ratings of movies by users. We randomly select 20% of the entries as a test set, and the remaining entries are split between a training set (80%) and a validation set (20%). For all the methods, we stop the regularization path as soon as the estimated rank exceeds $r_{max} = 100$. This is a practical consideration: given that the computations for high ranks demand more time and memory, we are interested in restricting ourselves to low rank solutions. Table 1 presents the results, averaged over 5 replications. For the MovieLens 100k dataset, HASI provides better NMAE than the other methods with a low rank solution. For the larger MovieLens 1M dataset, the precision, maximum number of iterations and maximum rank are decreased to $\epsilon = 10^{-6}$, $t_{max} = 100$ and $r_{max} = 30$. On this dataset, MMMF provides the best NMAE at maximum rank. HASI provides the second best performances with a slightly lower rank.

## 5   Conclusion

The proposed class of methods has shown to provide good results compared to several alternative low rank matrix completion methods. It provides a bridge between nuclear norm and rank regularization algorithms. Although the related optimization problem is not convex, experiments show that initializing the algorithm with the Soft-Impute algorithm of [4] provides very satisfactory results.

In this paper, we have focused on a gamma mixture of exponentials, as it leads to a simple and interpretable expression for the weights. It is however possible to generalize the results presented here by using a three parameter generalized inverse Gaussian prior distribution (see e.g. [19]) for the regularization parameters $\gamma_i$, thus offering an additional degree of freedom. Derivations of the weights are provided in the supplementary material. Additionally, it is possible to derive an EM algorithm for low rank matrix completion for binary matrices. Details are also provided in supplementary material.

While we focus on point estimation in this paper, it would be of interest to investigate a fully Bayesian approach and derive a Gibbs sampler or variational algorithm to approximate the posterior distribution, and compare to other full Bayesian approaches to matrix completion [20, 21].

**Acknowledgments**

F.C. acknowledges the support of the European Commission under the Marie Curie Intra-European Fellowship Programme. The contents reflect only the authors views and not the views of the European Commission.

## Footnotes

[1] Jester datasets can be downloaded from the url http://goldberg.berkeley.edu/jester-data/

[2]MovieLens datasets can be downloaded from the url http://www.grouplens.org/node/73.

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
