[Supplementary Material]

# Probabilistic Low-Rank Matrix Completion with Adaptive Spectral Regularization Algorithms: Supplementary Material

**Adrien Todeschini**
INRIA - IMB - Univ. Bordeaux
33405 Talence, France
Adrien.Todeschini@inria.fr

**François Caron**
Univ. Oxford, Dept. of Statistics
Oxford, OX1 3TG, UK
Caron@stats.ox.ac.uk

**Marie Chavent**
Univ. Bordeaux - IMB - INRIA
33000 Bordeaux, France
Marie.Chavent@u-bordeaux2.fr

## A  Estimation of the noise parameter $\sigma^2$

If we assume that $\sigma^2 \sim \mathrm{InvGamma}(a_\sigma, b_\sigma)$, then at each iteration of the algorithm we can maximize w.r.t. $\sigma^2$ given $Z^{(t)}$ in the E step to obtain

$$\sigma^{2(t)} = \frac{a_\sigma + ||X - Z^{(t)}||_F^2}{b_\sigma + mn}$$

## B  Proof of Eq. (13)

$$Q(Z, Z^*) = \mathbb{E}[\log(p(P_\Omega(X), P_\Omega^\perp(X), Z, \gamma))|Z^*, P_\Omega(X)]$$

$$= C_3 - \frac{1}{2\sigma^2}\mathbb{E}\left[\left\|P_\Omega(X) + P_\Omega^\perp(X) - Z\right\|_F^2 |Z^*, P_\Omega(X)\right] - \sum_{i=1}^r \mathbb{E}[\gamma_i|d_i^*]d_i$$

$$= C_3 - \frac{1}{2\sigma^2}\left\{\|P_\Omega(X) - P_\Omega(Z)\|_F^2\right.$$

$$\left. + \mathbb{E}\left[\left\|P_\Omega^\perp(X) - P_\Omega^\perp(Z)\right\|_F^2 |Z^*, P_\Omega(X)\right]\right\} - \sum_{i=1}^r \mathbb{E}[\gamma_i|d_i^*]d_i$$

$$= C_4 - \frac{1}{2\sigma^2}\left\{\|P_\Omega(X) - P_\Omega(Z)\|_F^2 + \left\|P_\Omega^\perp(Z^*) - P_\Omega^\perp(Z)\right\|_F^2\right\} - \sum_{i=1}^r \mathbb{E}[\gamma_i|d_i^*]d_i$$

$$= C_4 - \frac{1}{2\sigma^2}\left\{\left\|P_\Omega(X) + P_\Omega^\perp(Z^*) - Z\right\|_F^2\right\} - \sum_{i=1}^r \mathbb{E}[\gamma_i|d_i^*]d_i$$

## C  Generalization to other mixing distributions

Although we focused on a gamma mixing distribution for its simplicity, it is possible to use other mixing distributions $p(\gamma_i)$, such as inverse Gaussian or improper Jeffreys distributions. More generally, one can consider the three parameters generalized inverse Gaussian distribution [1], which includes the gamma, inverse gamma, inverse Gaussian and Jeffreys distributions as special cases. Table 1 provides the weights $\omega_i$ depending on the choice of $p(\gamma_i)$.

Table 1: Expressions of various mixing densities and associated weights. $K_\nu$ denotes the modified Bessel function of the third kind.

| Mixing density $p(\gamma_i)$ | Marginal density $p(d_i)$ | Weights $\omega_i = \mathbb{E}[\gamma_i \mid d_i^*]$ |
|---|---|---|
| $\text{Gamma}(\gamma_i; a, b) = \frac{b^a}{\Gamma(a)} \gamma_i^{a-1} e^{-b\gamma_i}$ | $\frac{ab^a}{(d_i+b)^{a+1}}$ | $\frac{a+1}{b+d_i^*}$ |
| $i\text{Gauss}(\gamma_i; \delta, \gamma)$ $= \frac{\delta}{\sqrt{2\pi}} e^{\delta\gamma} \gamma_i^{-3/2} e^{-\frac{1}{2}(\delta^2\gamma_i^{-1}+\gamma^2\gamma_i)}$ | $\frac{\delta}{\sqrt{\gamma^2+2d_i}} e^{\delta(\gamma-\sqrt{\gamma^2+2d_i})}$ | $\frac{\delta}{\sqrt{\gamma^2+2d_i^*}}\left(1 + \frac{1}{\delta\sqrt{\gamma^2+2d_i^*}}\right)$ |
| $\propto 1/\gamma_i$ | $\propto 1/d_i$ | $1/d_i^*$ |
| $\text{GiG}(\gamma_i; \nu, \delta, \gamma)$ $= \frac{(\gamma/\delta)^\nu}{2K_\nu(\delta\gamma)} \gamma_i^{\nu-1} e^{-\frac{1}{2}(\delta^2\gamma_i^{-1}+\gamma^2\gamma_i)}$ | $\frac{\delta\gamma^\nu}{K_\nu(\delta\gamma)} \frac{K_{\nu+1}\left(\delta\sqrt{\gamma^2+2d_i}\right)}{\left(\sqrt{\gamma^2+2d_i}\right)^{\nu+1}}$ | $\frac{\delta}{\sqrt{\gamma^2+2d_i^*}} \frac{K_{\nu+2}\left(\delta\sqrt{\gamma^2+2d_i^*}\right)}{K_{\nu+1}\left(\delta\sqrt{\gamma^2+2d_i^*}\right)}$ |

## D    Binary matrix completion

We have considered real valued matrices $X$. We now show how it is possible to apply the same methodology to binary, incomplete matrices of entries $Y_{ij} \in \{-1, 1\}$. Similarly to [2], we assume the following probit model

$$Y_{ij}|Z_{ij} \sim \text{Ber}\left(\Phi\left(\frac{Z_{ij}}{\sigma}\right)\right)$$

where $\Phi(x) = \int_{-\infty}^x \varphi(u)du$ is the cumulative distribution function of the standard Gaussian distribution with $\varphi(u) = \frac{1}{\sqrt{2\pi}}\exp(-\frac{u^2}{2})$. The model can be alternatively written using Gaussian latent variables $X_{ij}$

$$X_{ij}|Z_{ij} \sim \mathcal{N}(Z_{ij}, \sigma^2)$$
$$Y_{ij} = \begin{cases} +1 & \text{if } X_{ij} > 0 \\ -1 & \text{otherwise} \end{cases}$$

We will use the variables $X_{ij}$ as additional latent variables in the EM. We have

$$\mathbb{E}[X_{ij}|P_\Omega(Y), Z] = \begin{cases} Z_{ij} + \frac{\varphi\left(\frac{Z_{ij}}{\sigma}\right)}{1-\Phi\left(-\frac{Z_{ij}}{\sigma}\right)} & \text{if } Y_{i,j}^\Omega = +1 \\ Z_{ij} - \frac{\varphi\left(\frac{Z_{ij}}{\sigma}\right)}{\Phi\left(-\frac{Z_{ij}}{\sigma}\right)} & \text{if } Y_{i,j}^\Omega = -1 \\ Z_{ij} & \text{if } Y_{i,j}^\Omega = 0 \end{cases}$$

where we use the shorter notation $Y_{i,j}^\Omega = P_\Omega(Y)(i,j)$. We will now derive the EM algorithm, by using latent variables $\gamma_i$ and $X$. The E step is given by

$$Q(Z, Z^*) = \mathbb{E}[\log(p(P_\Omega(Y), X, Z, \gamma))|Z^*, P_\Omega(Y)]$$
$$= C_5 - \frac{1}{2\sigma^2}\|X^* - Z\|_F^2 - \sum_{i=1}^r \mathbb{E}[\gamma_i|d_i^*]d_i \qquad (1)$$

where the matrix $X^*$ is defined as

$$
X_{ij}^* = \begin{cases}
Z_{ij}^* + \dfrac{\varphi\left(\frac{Z_{ij}^*}{\sigma}\right)}{1-\Phi\left(-\frac{Z_{ij}^*}{\sigma}\right)} & \text{if } Y_{i,j}^{\Omega} = +1 \\[2em]
Z_{ij}^* - \dfrac{\varphi\left(\frac{Z_{ij}^*}{\sigma}\right)}{\Phi\left(-\frac{Z_{ij}^*}{\sigma}\right)} & \text{if } Y_{i,j}^{\Omega} = -1 \\[2em]
Z_{ij}^* & \text{if } Y_{i,j}^{\Omega} = 0
\end{cases}
$$

Again, the maximum of the function (1) is obtained analytically using a weighted soft thresholded SVD on the matrix $X^*$.