[Reviews · NeurIPS 2013]

Submitted by Assigned_Reviewer_5

SUMMARY

This paper studies the problem of low rank matrix completion which exists in many real-world applications such as collaborative filtering for recommender systems. A previous work (ref [4]) proposed a scalable algorithm called Soft-Impute for solving a convex optimization problem involving the nuclear norm as a regularizer. Like previous work such as probabilistic matrix factorization (PMF), this paper gives the problem a probabilistic interpretation by relating the (non-probabilistic) optimization problem to a MAP estimation problem. Different (concave) penalty functions of the nuclear norm are proposed and then an EM algorithm is proposed to solve the MAP estimation problem. The algorithms proposed in this paper are more general than the Soft-Impute algorithm proposed in [4] in that the latter comes as a particular case.

QUALITY

The main contribution of this paper is that it proposes new penalty functions based on the nuclear norm for low rank matrix completion. Although experimental validation has shown that the method (HASI) that makes use of the proposed penalty functions generally outperforms other methods compared, its technical soundness could be more strongly articulated via theoretical analysis. This is especially the case when the proposed regularizers are non-convex and hence make the optimization problem more challenging than solving a convex optimization problem. This price would be worth paying only if it had advantages preferably supported well by theoretical analysis, besides experimental validation. Even on the experimental validation, besides showing the results as in Table 1, the reader could gain a deeper understanding of the strengths and weaknesses of different methods if more detailed analysis of the results could also be provided. For example, we only know that HASI usually gives the lowest NMAE except for the MovieLens 1M dataset (which comes second), but we have no clue on what makes the performance degrade in this case. Is it because of the large dataset? Does the proposed method perform less satisfactorily for large datasets due to non-convexity of the optimization problem? Also, there is relatively large variation in the rank among different methods. It would help if the authors could comment on this aspect as well.

Since the proposed method is based on point estimation, calling it “Bayesian” may be controversial. It is more appropriate to refer to it as “probabilistic” instead. Related previous work also used this convention. For example, for matrix factorization/completion, a point estimation method is called probabilistic matrix factorization:

R. Salakhutdinov and A. Mnih. Probabilistic matrix factorization. NIPS, 2008.

while its full Bayesian extension is called Bayesian probabilistic matrix factorization:

R. Salakhutdinov and A. Mnih. Bayesian probabilistic matrix factorization using Markov chain Monte Carlo. ICML, 2008.

Nevertheless, this is a well-written paper which is a pleasure to read. I also like the supplementary material which includes discussions on generalization to other mixing distributions as well as the binary case which does occur in practice.

CLARITY

The paper is quite clearly written, making it easy for the reader to follow. I particularly like the organization which presents the complete matrix case first and then considers the more general, incomplete matrix case for matrix completion which is the focus of this paper.

Nevertheless, I have a few minor comments. Addressing them will further improve the clarity of the paper.

Line 37:
“... the low rank assumption is sensitive ...” Do you actually mean “sensible”? I don’t understand why it is “sensitive”.

In Eq (7), strictly speaking some constant terms are missing even though they have no effect on the penalty function for optimization.

In Figure 3, the labels on the axes are too small to be visible without zooming in the PDF file.

Line 184:
When you refer to \gamma as regularization parameters, it would help to point to the specific regularized objective function in which they play the role of regularization parameters. It may not be clear to the reader.

In Eq (13), it should be the sum of two terms, not their difference.

ORIGINALITY

Needless to say, matrix completion is not a new problem. It has already aroused a lot of research interests over the past decade. Also, quite a few models have been given a probabilistic interpretation. So, this paper is not novel in this aspect either. The main contribution of this paper is on the introduction of some penalty functions based on the nuclear norm. As far as I know they have not been introduced in the context of matrix factorization/completion and related problems.

In fact, many of the techniques used in the paper are existing methods and results too. For example, the global optimality of the M-step of the EM algorithm in Eq. (9) is guaranteed by previous results reported in [11, 12], not this work; the scalability is due to the PROPACK algorithm which is from previous work [4]. If one has to comment on the novelty of this paper, it would be on the combination of these techniques and findings from previous work in a coherent way.

SIGNIFICANCE

The experimental results reported seem quite good, but it is premature to draw conclusions on the practical significance of the proposed method based solely on these relatively small-scale experiments. Its significance could be articulated more convincingly by making extensions at least along the following directions:
- Using larger datasets, e.g., Netflix dataset
- Empirical comparison with more state-of-the-art methods, e.g., Bayesian methods, nonlinear (kernel) methods, deep learning methods
- Theoretical analysis of new penalty functions
Summary: This paper proposes new penalty functions based on the nuclear norm for low rank matrix completion. It is in general technically sound and is clearly written, making it a pleasure to read. The significance of the proposed method could be articulated more convincingly via both theoretical analysis and more detailed analysis of the empirical findings.

Submitted by Assigned_Reviewer_6

This paper proposes a new penalty, called Hierarchical Adaptive Nuclear Norm, for matrix completion problems. An EM algorithm is derived for optimization. The advantage of new prior is that it includes many other interesting penalties, and also that the implementation is simple (based on SVD).

Quality of this paper is good. The paper is clear, but figure quality can be improved (increase the font size and modify the figure so that lines for different methods are shown clearly). The contributions made are original and significant.

Also, Eq. 7, please include the '+ cnst'.
Summary: The new penalty is interesting and useful, and this could help design better probabilistic models.

Submitted by Assigned_Reviewer_7

The paper presents a flexible extension to the nuclear norm regularization that bridges the gap between the rank-regularization and the plain nuclear norm regularization. The idea is based on a probabilistic interpretation of nuclear norm regularization and a straightforward Bayesian extension to it. The extended nuclear norm regularization is adaptive as it penalizes higher singular values less heavily. Matrix approximation with this regularization can be solved iteratively via a EM-style algorithm.

This is a very good paper. It's very well written and very easy to follow. The idea presented here is very simple and well-known, yet very enlightening. The approach proposed is also very interesting and useful. Overall, I think this paper has potentially good impact to matrix approximation as well as trace-norm regularization in general.

My only concern is that the conclusion of this paper seems to contradict somehow with that of another paper in the literature,

Salakhutdinov and Srebro: Collaborative Filtering in a Non-Uniform World: Learning with the Weighted Trace Norm.

Minor: (7) is not correct, the two sides of the equation equal up to a constant.
Summary: The paper presents a flexible extension to the nuclear norm, the idea is novel and the algorithm is very useful.
Author Feedback

Author rebuttal: We thank the reviewers for their positive and constructive feedbacks.

We agree that the term "Probabilistic" is more appropriate than "Bayesian", and we will modify it in the revised version.

The results tend generally to be less good for our method as the matrix becomes sparser. The algorithm then takes a longer number of steps to converge and is more likely to get trapped in local optima. Nonetheless, to obtain fair comparisons with alternative methods, we used a very simple setting for the EM algorithm, with a single initialization based on Soft-Impute. Multiple initializations, or the use of the various acceleration techniques of the EM algorithm could be used to improve the results.

Concerning the lack of theoretical analysis, the non convexity of the objective makes the problem very challenging to analyse, and we leave it as an open problem. Note that one can resort to classical results on the EM (Wu, 1983) to assess the convergence of the algorithm to a local maximum.

Concerning the ranks obtained on the real datasets (table 1), we observe that, on the one hand, MMMF and Soft-Impute methods have unimodal behavior (see figure 6) and generally provide the best test error at high rank solutions. Soft-Impute+ also has unimodal behavior but generally has its minimum at a lower rank than Soft-Impute. On the other hand, Hard-Impute and HASI exhibit a bimodal behaviour and thus can yield best test error either at a low rank or a high rank.

Regarding the conclusions of Salakhutdinov and Srebro, their algorithm regularizes users and movies factors. Our construction regularizes the singular values of the matrix.
We agree nonetheless that the conclusion of the paper of Salakhutdinov and Srebro are thought provoking and they seem to be in contradiction with the use of Bayesian/probabilistic approaches.